# Oral Motor Impairments Contribute to Weight Status of Adults with Severe Cerebral Palsy

**DOI:** 10.3390/nu15245042

**Published:** 2023-12-08

**Authors:** Aslak Emil Lyster, Solvejg Lis Hansen, Christina Therese Andersen, Jens Bo Nielsen, Klaas Westerterp, Loek Wouters, Bente Kiens, Anina Ritterband-Rosenbaum

**Affiliations:** 1The August Krogh Section for Molecular Physiology, Department of Nutrition, Exercise and Sports, University of Copenhagen, Universitetsparken 13, 2100 Copenhagen, Denmark; ael@nexs.ku.dk (A.E.L.); bkiens@nexs.ku.dk (B.K.); 2Elsass Foundation, Holmegaardsvej 28, 2920 Charlottelund, Denmark; svjh@novonordisk.com (S.L.H.); ca@elsassfonden.dk (C.T.A.); jbnielsen@sund.ku.dk (J.B.N.); 3Department of Neuroscience, University of Copenhagen, Blegdamsvej 3, 2200 Copenhagen, Denmark; 4Department of Nutrition and Movement Sciences, The Maastricht University, P.O. Box 616, 6200 MD Maastricht, The Netherlands; k.westerterp@maastrichtuniversity.nl (K.W.); l.wouters@maastrichtuniversity.nl (L.W.)

**Keywords:** body weight, dietary registration, cerebral palsy, oral motor impairment, metabolic health status, double-labeled water, resting metabolic rate, energy balance

## Abstract

Adults with severe cerebral palsy (CP) are susceptible to malnutrition and metabolic disorders due to limited daily physical activity and challenges related to eating. We hypothesized that the condition of being underweight arises from inadequate energy intake due to difficulties in eating, rather than heightened total energy expenditure or an elevated resting metabolic rate. The present study encompassed 17 adults with severe CP (classified as GMFSC III–V). Energy intake, utilization, and expenditure were gauged via thorough dietary recordings and double-labeled water (DLW) analyses. Resting metabolic rates were assessed through indirect calorimetry, and metabolic health was investigated via blood samples. Oral motor function, eating assessment during meals, and weight fluctuations throughout the experimental period were also evaluated. We found significant correlations between weight, oral impairments (*p* < 0.01), and eating difficulties (*p* < 0.05). While total energy expenditure and daily consumption were similar between underweight (UW) and overweight (OW) individuals, significant variability in both expenditure and intake was evident within the UW group. Particularly, those with lower BMIs experienced heightened mealtime impairments and complications. Our present findings indicate that eating difficulties are the central concern for UW status in this population.

## 1. Introduction

Cerebral palsy (CP) results from disturbances in neural development occurring before or around birth [1] and affects approximately 2–3 out of 1000 newborns, resulting in an estimated global prevalence of 17 million affected individuals [2].

Although CP is typically regarded as a childhood motor disability [3,4,5], its impact on various aspects of life persists throughout adulthood, influencing motor function, cognitive abilities, and mental health [2,6,7,8].

In the adult CP population, the majority fall within the mild-to-moderate range (I–III) of the Gross Motor Function Classification System (GMFCS), while only about one third are classified as more severely affected (IV and V), necessitating additional care, such as living in caretaker-home facilities [9].

Physical and metabolic health conditions in adults with CP are a subject of significant concern [2,6,7,8,10,11,12,13]. For many, a sedentary lifestyle may be linked to an increased risk of metabolic diseases [2,6,7,8], and for others, eating difficulties may lead to malnutrition and being underweight (UW) [10,11,12,13].

Recent studies have indicated that children with severe CP (GMFCS IV–V) tend to have lower body weight compared to children with mild CP (GMFCS I–II) and typically developed children. Many of the children with severe CP are categorized as UW (BMI < 18.5) [14]. A similar high prevalence of being UW is observed in adults with CP (GMFCS I–V), with 28–34% of this population identified as UW [10,11,12,13].

The concept of energy balance, which involves the net effect of energy consumption and total energy expenditure, plays a crucial role in understanding these weight differences. Total energy expenditure includes resting metabolic rate, the thermic effect of food, growth, and physical activity level. Evidence suggests that the prevalence of being UW (BMI < 20) in children and adults with cognitive and movement disabilities, including CP, may be related to swallowing difficulties and lower energy consumption when dependent on a caregiver during eating situations [12,15,16].

However, conflicting findings exist regarding the total energy expenditure of adults with CP, particularly in terms of daily physical activity and resting metabolic rate. A study has reported elevated resting metabolic rates in adults with CP compared to neurologically intact controls when differences in fat-free mass were adjusted for [17]. Conversely, other studies have demonstrated reduced resting metabolic rates in people with CP when compared to age-matched, neurologically intact controls, although these investigations were primarily conducted with children and young adults with varying severity [18,19].

To elucidate the underlying reasons for low body weight in adults with severe CP (GMFCS III–V), this study was initiated with a comprehensive test battery aimed at investigating energy balance and identifying the main mechanisms contributing to poor body-weight status. Additionally, this study seeks to provide evidence of the metabolic consequences of being an UW person with severe CP (GMFCS III–V). The hypothesis posits that the UW condition in this group may be attributed to a low energy intake resulting from eating difficulties rather than a high total energy expenditure or an elevated resting metabolic rate.

## 2. Materials and Methods

### 2.1. Ethics Approval and Consent

All subjects in the study received written and spoken information about the study and provided signed informed consent in the presence of their primary caregivers in the caretaker home. The study was approved by the Copenhagen Ethics Committee (protocol number: H-16028528) on 26 October 2018 and was performed in accordance with the Helsinki Declaration. All personal data were anonymized. Names were assigned to an identification number, and data were stored according to the guidelines provided by the Danish Data Protection Agency.

### 2.2. Subjects and Study Design

Out of 35 eligible adult subjects diagnosed with severe CP based on their medical records, 18 subjects agreed to participate in the study. Subjects were recruited from a 24 h caretaker home in the greater area of Copenhagen, Denmark. A total of 17 adults (average age: 48.9 ± 15.6 years, 13 men/4 women) completed the trial, as one participant withdrew from the study at an early stage due to personal reasons. Participants had GMFCS scores between levels III and V [20], which were evaluated by a physical therapist. The GMFCS is a 5-level clinical classification system that describes the gross motor function of people with CP on the basis of self-initiated movement abilities [20]. Subjects were separated into three groups based on body weight: underweight (UW), BMI < 20; normal weight (NW), BMI = 20–25; and overweight (OW), BMI > 25. There was no statistically significant difference in age or height between groups. Total body weight was 26% lower in the UW group compared with the NW group (*p* < 0.05), while BW was 19% higher in the OW group compared with the NW group (*p* = 0.065). A total of 60% of the UW participants and 50% of the NW ones were diagnosed with dyskinetic CP, while all OW participants were diagnosed with spastic CP (Table 1).

The study was carried out from August 2020 to January 2021, and all subjects were encouraged to continue their regular work and leisure activities, and they and their supporting contact persons at the caretaker home were also instructed not to change their regular dietary plan throughout the whole study period.

The design of the study included different tests that investigated the subjects’ energy balances. Information was obtained about energy intake based on weighed dietary recordings and occupational therapeutic mealtimes combined with oral motor-function evaluations. Those two tests were completed at different time points during the project and did not overlap with each other to make sure full attention was given to the different tests. In addition, resting metabolic rate was measured using indirect calorimetry, and double-labeled water (DLW) was applied to measure energy expenditure and body composition. Furthermore, blood sampling and anthropometric measures were performed to evaluate subjects’ metabolic status.

Figure 1 shows an overview of the test battery used in the study protocol, which included measurements for the energy balance and metabolic status of the group. Numbers in parentheses indicate the number of participants who were able to perform tests.

All subjects were able to complete all tests, except for one subject who was not able to perform the indirect calorimetry test (not feeling comfortable using the mask needed for the experiment) and one subject who was ill (not related to participation in the study) when the blood sample was collected.

### 2.3. Energy Intake—Weighed Dietary Recording

Subjects conducted dietary recording, in which all food and beverages were weighed within 1 g of accuracy over five weekdays within one week. Caregivers and kitchen personnel weighed and recorded food intake of subjects, as they were unable to perform the tasks unassisted. Recording was done with specialized food lists adapted to the project, and all involved personnel were instructed on how to record all consumed food items using kitchen weights. Any spills or leftovers were also measured and recorded to obtain accurate dietary details. A representative from the research team, trained in the method, was present during most of the meals (breakfast/lunch/dinner) to advise, assist, and observe the proceedings. The weighed dietary recording was considered acceptable, with a minimum of four days with adequate recordings. All subjects were encouraged to maintain their regular eating habits during the assessment of their diets. Dietary records were analyzed using the online program Vitakost (Vitakost, Kolding, Denmark) to quantify macronutrients, micronutrients, vitamins, and general energy intake.

### 2.4. Energy Intake—Mealtime and Oral Motor-Function Evaluations

The subjects’ oral motor functions during a mealtime (breakfast, lunch, or dinner) were evaluated by a professional occupational therapist based on an eating session which represented a normal mealtime event. The session was video-recorded to be evaluated afterwards. The evaluation focused on body position, chewing, maneuvering food in the mouth, swallowing, drooling, food residue, swallowing errors, and the length of the mealtime. During the evaluation, the subjects were classified with the Eating and Drinking Ability Classification System (EDACS) [21]. Furthermore, various motor-control exercises were performed to test the motor function of the mouth and tongue. Examples of functions include stretching the tongue out towards the chin and nose, stretching the tongue towards the right and the left outside the mouth, and pronunciation of the sounds ‘N’, ‘G’, and ‘NG’. The scores given in the evaluation were as follows: significantly affected = 3, very affected = 2, slightly affected = 1, or not affected = 0. The number of observed swallowing errors and number of coughs were classified as either more than 4, between 2 and 4, 1, or 0. Summation of the individual scores for eating and drinking complications and oral motor-impairment evaluations were used for further analyses [22].

### 2.5. Energy Expenditure—Resting Metabolic Rate (RMR)

Resting metabolic rate (RMR) was measured with indirect calorimetry using an Oro-Nasal Mask (Hans Rudolph, Kansas City, KS, USA) connected with an online gas analyzer system (Masterscreen CPX SBx, Carefusion, Wurmlingen, Germany). Prior to testing, subjects were familiarized with the Oro-Nasal Mask. All measurements were conducted in an undisturbed thermoneutral room that was located at the caretaker home and well known to the subjects and in the presence of their personal caregiver. On the day of the test, the subjects arrived early in the morning in a fasted state (>10 h after last food intake). Subjects rested for 20 min, which was followed by two 10-min indirect calorimetry measurements interspaced by a 2 min break [23]. During all procedures, subjects were seated. Movements during indirect calorimetry were noted by the operator. For data analysis, the first five minutes were discarded, as that period corresponded to a habituation period to the mask, and the interval of at least four minutes with the lowest coefficient of variation was selected [23]. Resting metabolic rate was calculated using the Weir equation [24]:Resting metabolic rate (kcal/day) = (3.94 × VO_2_ (L/min) + 1.11 × VCO_2_ (L/min)) × 1440 (1)

All subjects were able to complete the resting metabolic rate measurements, except one. For this subject, basal metabolic rate was calculated using the revised Harris–Benedict equation [25]:Basal metabolic rate (kcal/day) = 88.362 + (13.397 × weight (kg)) + (4.799 × height (cm)) − (5.677 × age (years))(2)

### 2.6. Energy Expenditure—Indirect Body Composition via Double-Labeled Water (DLW)

Fat-free mass was determined using the skinfold thickness measurement technique (Harpenden Skinfold Caliper) to be used for the DLW procedure (see below). Since most of the subjects were unable to stand upright, the skinfold measurements were performed in a supine position. Four skin areas were chosen bilaterally: subscapular, biceps, triceps, and suprailiac. All areas were measured three times, and an average measure was noted for all skinfold measurements and used to determine fat-free mass [26]. All measures were conducted by the same operator.

The DLW method for the measurement of energy expenditure is an innovative variant on indirect calorimetry. It is based on the discovery that oxygen in respiratory carbon dioxide is in isotopic equilibrium with the oxygen in body water. The technique involves enriching the body water with an isotope of oxygen and an isotope of hydrogen and then determining the washout kinetics of both isotopes. Most of the oxygen isotope is lost as water, but some is also lost as carbon dioxide because CO_2_ in body fluids is in isotopic equilibrium with body water due to exchange in the bicarbonate pools. The hydrogen isotope is lost as water only. Thus, the washout for the oxygen isotope is faster than the hydrogen isotope, and the difference represents the CO_2_ production. Total daily energy expenditure was measured with the DLW method over a 14-day period under free-living conditions using the “six-point” protocol described by Westerterp and colleagues [27]. A background urine sample was collected before the ingestion of the individualized doses of DLW. The individual doses, which were based on the fat-free masses measured with the skinfold method, were consumed as the last meal of the day (sample 0). Both urine collections and consumption of DLW were supervised by trained therapists. To accommodate the swallowing challenges for some of the subjects with ingestion of DLW, a thickening product was added to the DLW. Two urine samples were collected the following day (sample 1 and 2), seven days after (sample 3 and 4), and fourteen days after (sample 5 and 6) administration of DLW. The samples were collected in the morning and in the evening and stored in a freezer at −18 °C. For subjects using catheters or diapers, special care was given to ensure the correct timing of the samples.

### 2.7. Metabolic Status—Blood Test

A blood sample of 16 mL was collected from an antecubital vein in the fasted state (>10 h after a meal) in the morning by trained personnel, whereafter subjects rested for a short period of time under observation by the personnel. A total of 10 mL was used for blood serum analysis and 6 mL of blood was centrifuged for plasma analysis. Plasma was stored at −20 °C until subsequent analysis.

### 2.8. Anthropometric Measures

During the study, body weight was measured, specifically for skinfold measurements, weighed dietary registration, and the DLW method, according to testing procedures. This was performed in the morning, in a fasted state and after using the toilet. A digital sling scale and a lift sheet (Guldmann A/S, Graham Bells Vej 21-23A, Aarhus N, Denmark) were used for all body weight measurements. Body weight measurements undertaken for a particular test were subsequently employed in the data analysis.

### 2.9. Analysis and Statistics

#### 2.9.1. DLW Analysis

Urine samples were analyzed at NUTRIM Research Institute, University of Maastricht, Netherlands, using gas isotope ratio mass spectrometry. Isotopic enrichment of the post-dose urine samples was analyzed relative to background enrichment. Carbon dioxide production rate was calculated based on newly derived equations [28]. Total body water was estimated from the DLW analysis and used to calculate fat-free mass (Equation (3)):(3)$$FatFreeMass\;(kg)=\frac{TotalBodyWater\;(kg)}{0.73}$$
(4)$$TotalFatMass\;\left(kg\right)=TotalBodyWeight\;\left(kg\right)-FatFreeMass\;(kg)$$

#### 2.9.2. Blood Analysis

For the blood samples, plasma glucose concentrations were measured with an ABL800 flex (Radiometer, Copenhagen, Denmark) and insulin concentrations were measured with an enzyme-linked immunosorbent assay (ELISA) (ALPCO insulin ELISA, Salem, MA, USA). Plasma concentrations of Brain-Derived Neurotrophic Factor (BDNF) were measured using an ELISA (Cat. No. CYT306, Chemicon International Chemokine from Millipore Corporation, MA, USA). Plasma 25 (OH) vitamin D levels were measured using an ELISA kit (ab213966, 25 (OH) Vitamin D ELISA kit, Abcam, Amsterdam, the Netherlands). Total, HDL, and LDL cholesterol and triacylglycerol plasma concentrations were measured with an enzymatic colorimetric method (Hitachi 912 automatic analyzer, Boehringer, Ingelheim, Germany).

The homeostasis model assessment–insulin resistance index (HOMA-IR) was used to estimate insulin resistance in subjects and was calculated with the following equation (Matthews et al., 1985):(5)$$HOMA-IR=\frac{Fasting\;insulin\;concentration\;\left[\frac{\mu U}{mL}\right]\;*\;Fasting\;glucose\;concentration\;[\frac{mmol}{L}]}{22.5}$$

#### 2.9.3. Statistics

Graphpad Prism 9 (GraphPad Software, 2365 Northside Dr., Suite 560, San Diego, CA, USA) was used for all statistical calculations, which included correlations with a Pearson correlation analysis and one-way ANOVA with a Tukey post hoc test. We included a chi^2^ test examining statistical differences within clinical parameters for the three groups. All results were calculated as the average for the subgroups, and graphs are shown with 1 standard deviation (SD). All personal characteristics are presented in tables with averages and ± 1SD. The remaining data are presented as means with standard error of mean (SEM). The level of significance was set to 0.05.

## 3. Results

### 3.1. Results of the Groups

In Table 1, we present the subject characteristics of the different groups. Data from the chi^2^ tests for detecting the statistical frequencies of GMFCS, type of CP, and topographical classifications between the groups showed no differences, with *p*-values of 0.998, 0.948, and 0.988, respectively.

### 3.2. The Impact of Oral Motor Impairments on Weight Status

The BMI of adults with severe CP was negatively correlated with oral motor impairments (R^2^ = 0.51, *p* < 0.01, Figure 2A) and eating and drinking complications (R^2^ = 0.4, *p* < 0.05, Figure 2B). A lower BMI was correlated with a higher degree of both oral motor impairments (Figure 2A, *p* < 0.01) and eating and drinking complications (Figure 2B, *p* < 0.01). In line with this finding, the UW group, on average, spent approximately twice as long as the OW group at mealtimes (27.4 ± 6.1 min vs. 16.3 ± 2.8 min) (Table 2). Totals of 80% and 50% of the UW and NW groups, respectively, scored EDACS level IV compared to only 12.5% in the OW group. In the UW and NW groups, totals of 80% and 75%, respectively, were dependent on their caregivers in eating situations. In the OW group, 37.5% were totally dependent on their caregivers and 50% could eat independently of caregivers (Table 2).

### 3.3. Energy Expenditure and Energy Balance in Adults with Severe CP

Total daily energy expenditures (kcal/day) were on average 1748 ± 357, 1869 ± 221, and 1753 ± 400 kcal/day in the UW, NW, and OW groups, respectively. There was no difference between groups (Figure 3A, *p* = 0.85). However, large variations in total energy expenditure were observed in the UW (range: 1442–2216 kcal/day) and OW groups (range: 1366–2367 kcal/day). Daily energy consumption in the three groups reached 2055 ± 706 kcal/day in the UW group, 1750 ± 395 in the NW group, and 1692 ± 280 in the OW group, and no statistical difference could be detected between groups (Figure 3B, *p* = 0.40). The UW group had a net positive energy balance of 307 ± 433 kcal/day on average, while both the NW and OW groups had net negative energy balance (Figure 3C).

The fat-free mass calculated using the DLW method was not significantly different between groups (UW: 38.1 ± 4.8 kg (range: 34.4–44.2), NW: 44.7 ± 6.5 kg (range: 35.5–50.6), OW: 43.4 ± 4.9 kg (range: 35.6–51.2), *p* = 0.15). The body fat percentage was significantly higher in the OW group compared to the UW group (41.7 ± 9.6 vs. 17.6 ± 6.4%, *p* < 0.05), while there was no significant difference between the UW and NW groups (17.6 ± 6.4 vs. 29.7 ± 4.8%, *p* = 0.09).

Resting metabolic rate related to fat-free mass (kcal/kg fat-free mass/day) did not reach significant differences between the groups (UW: 31.0 ± 2.0, NW: 31.5 ± 5.6, OW: 27.4 ± 6.5 kcal/kg FFM/day, *p* = 0.09). RMR constituted a total of 72% of total energy expenditure (range: 56–87% of total energy expenditure) on average for all subjects.

Body weight changed 0.6 ± 1.2 kg on average from the first to the finale body weight measurement in the study period for each subject, and there was no difference in body weight changes between groups (UW: 0.2 ± 1.0, NW: 0.8 ± 1.8, OW: 0.8 ± 1.2, p = 0.66).

### 3.4. Diet

Dietary protein intake was 70 ± 17 g/day on average in all groups (Table 3). However, the intake of protein related to body weight (g/kg/day) was significantly higher in the UW group compared to both the NW (1.58 ± 0.2 vs. 1.03 ± 0.2 g/kg/day, *p* = 0.055) and OW groups (1.58 ± 0.2 vs. 0.95 ± 0.1 g/kg/day, *p* < 0.05) (Table 3).

The UW group consumed 47 ± 5.6 E% from fat, which was 12 E% higher than the OW group (*p* < 0.05) and 11 E% higher than the NW group (*p* = 0.12) (Table 3).

Carbohydrate intake in the UW group was 38.1 ± 4.0 E%, which was significantly lower than the intake in the NW group (50 ± 2.6 E%, *p* < 0.05, Table 3). However, the total amount of carbohydrate intake (g/day) per day did not differ between groups. We did not find a statistically significant difference between the UW and OW groups (*p* = 0.07).

The daily intake of added sugar in the NW group constituted 22 ± 2.7 E% of total energy consumption in comparison to 11 ± 2.3 E% and 10 ± 2.9 E% in the UW and OW groups, respectively (Table 3).

In the OW group, the dietary fiber intake of 2.7 ± 0.1 g/MJ was almost twice the amount of the UW and NW groups, which had 1.5 ± 0.1 (*p* < 0.001) and 1.5 ± 0.2 g/MJ (*p* < 0.001), respectively (Table 3).

A total of 80% of the UW group took a standardized vitamin supplement, which was the case for only 50% in the NW group and 25% in the OW group. Moreover, 75% of the subjects in the NW group and 25% in the OW group, but none in the UW group, took vitamin D supplements. Large variations were observed in plasma 25-(OH) vitamin D concentrations in all groups. (Table 4).

### 3.5. Metabolic Status

Plasma glucose concentrations were 5.3 ± 0.2, 4.8 ± 0.1, and 5.6 ± 0.3 mmol/L in the UW, NW, and OW groups, respectively. There were no significant differences between the groups (Table 4).

The plasma insulin concentration was 12.4 ± 2.3 µU/mL in the OW group, which was twice as high as in the UW and NW groups (Table 4). The HOMA-IR scores averaged 3.1 ± 0.7 in the OW group, with a range between 1.3 and 6.5, and five out of seven subjects had HOMA-IR scores above 2.5, compared with 1.5 ± 0.3 in both the UW and NW groups (Table 4).

Large variations were obtained regarding plasma HDL, LDL, and total cholesterol concentrations in all groups (Table 4).

The plasma BDNF concentration was 4407 ± 1303 pg/mL in the OW group, which was significantly lower compared with the UW (*p* < 0.05) and NW groups (*p* = 0.07) (Table 4).

## 4. Discussion

This study explores whether energy expenditure, energy intake, and metabolic risk factors are linked to weight and oral motor functions in adults with severe CP. We observed a significant inverse correlation between oral motor functions, eating complications, and BMI. Elevated EDACS scores indicated a heightened risk of being UW due to issues with oral manipulation, swallowing, drooling, and choking/coughing. Paradoxically, UW individuals displayed higher energy intake than expenditure, possibly attributed to the caregiver-assisted eating of all UW subjects, which contrasted with partial assistance for the OW and NW groups (50% and 75%, respectively). The majority of the UW subjects had modified food textures (80% vs. 62.5% in the OW group and 75% in the NW group) and prolonged eating times (Table 2), influencing their elevated energy intake. This underlines the vital role of healthcare personnel in adapting diets to address oral motor challenges, as suggested by Humphries et al. (2009). The present longitudinal study spanned four months, suggesting that the increased energy intake should ideally raise UW individuals’ body weight; however, their weight remained unchanged throughout the study period.

We acknowledge the inherent limitations of weighed dietary recording as a method for quantifying energy intake, as it relies on various uncertainties surrounding food measurement and assessment of leftovers. A recent study [30] demonstrated that the energy intake of adults aged 18–60 was underestimated by 16% in a 7-day food recording. In the present study, a five-day weighed dietary recording was used, meticulously weighing all food and beverages to the nearest gram. The caretaker-home staff underwent training for dietary recording and adhered to leftover measurement guidelines. Measuring nutritional remnants, particularly in the UW group with the highest EDACS score, posed challenges due to coughing and drooling, potentially leading to inaccuracies even when collecting bib-trapped leftovers. Such discrepancies likely contributed to the UW group’s lack of weight gain during the experiment. Moreover, the five-day dietary records were confined to weekdays, which potentially results in an error in daily intake on a weekly basis. The risks of missing weekend snacking for some subjects or registering a lower intake for others are present and could potentially explain why the UW group did not gain weight during the period of the study despite the higher intake. This might further be supported by the fact that nobody in the UW group was able to eat independently of their caretaker, while 50% of the OW group were independent of the caretakers in eating situations.

The challenges of meeting daily energy requirements and heightened risks of malnutrition among individuals with severe intellectual and developmental disabilities, including those with CP, are well known [7]. Prior to our study, the caretaker home had, therefore, already taken proactive measures to address this concern, aiming to rectify low body weight through dietary adjustments, improvements in eating conditions, and food-consistency modifications. This is consistent with our finding of a significantly elevated intake of dietary fat (Table 3) within the UW group compared to the other groups. This aligns with earlier recommendations to increase dietary energy intake for adults with CP [16]. Furthermore, food textures and drink consistencies were tailored to individual needs, and caregivers, along with occupational therapists, provided assistance during mealtimes to counteract or prevent low energy intake. This intervention potentially helped individuals sustain adequate energy intake by enhancing eating safety and alleviating fatigue, thus reducing the risks of coughing and choking. While limited research has explored the correlation between diet modifications in terms of texture and consistency [15,31] and energy consumption, we argue that an enhanced emphasis on eating safety may impact energy intake or, at the very least, contribute to an improved mealtime experience.

The UW group achieved an average daily intake of approximately 2000 kcal. This finding is unexpectedly high compared to the other groups and contradicts previous observations of compromised energy consumption among individuals requiring eating assistance, experiencing extended meal durations, or exhibiting significant oral motor impairments [10,12,16]. We contend that the notable energy intake can be attributed to the subjects’ residence in a 24 h caretaker facility that is staffed with well-trained personnel who offer mealtime support, modify food and beverage textures, and provide supplemental macronutrient intake.

In the present cohort of adults with severe CP, the mean total daily energy expenditure was 1798 ± 68 kcal/day for the entire group, and the total daily energy expenditure did not exhibit significant intergroup differences. The generally low total daily energy expenditure and interindividual variations observed in this cohort with severe CP, irrespective of weight status, align with the findings of Johnson and colleagues (1997) with non-ambulatory adults with CP (GMFCS: I-V). The total daily energy expenditure was computed using the DLW method, which is considered the gold standard for measuring energy expenditure. However, using the DLW method might pose potential challenges in a population prone to swallowing complications. To ensure accurate DLW ingestion and minimize spillage, the caretaker home’s occupational therapist devised customized procedures for each subject, necessitating extensive staff training. Prior reports highlight considerable variability in total daily energy expenditure among adults with CP using the DLW method [32]. Although their energy intake remained similarly low, subjects in the present study maintained energy balance. Promoting energy intake could entail elevating activity levels to establish a positive energy balance spiral. Despite substantial variability in total daily energy expenditure among adults with severe CP, we argue that weight status is not closely tied to total daily energy expenditure, given the absence of evidence indicating a higher energy expenditure in the UW group relative to the other groups. It could be speculated that a more severe type of CP (CP combined with dyskinetic movements) may impact weight status or energy expenditure, as we observed a higher prevalence in the UW and NW groups compared to the OW group, which consisted only of individuals with spastic CP (Table 1). However, this should be taken with caution due to the low number of subjects with dyskinetic CP in the study, and more research is needed.

Overall, our findings regarding energy balance across all the groups suggest that participants’ prevailing weight status may stem from prior energy imbalances spanning childhood through adulthood, possibly influenced by documented oral motor impairments and compromised energy intake [10,14]. We are aware that due to the relatively low number of participants in the present study, we cannot expand the findings to the whole population of severely affected adults with CP. On the other hand, we have included several different techniques to understand energy intake and expenditure combined with metabolic data, which is a great advantage for understanding energy balance.

Insufficient energy intake can lead to inadequate micronutrient and essential-fat consumption, necessitating food with varied, highly nutritional content to mitigate metabolic risks. Among 17 participants, 13 had low energy intake (~2000 kcal/day), elevating the risk of micronutrient insufficiency [29]. Our data reveal that, on average, severely UW adults with CP had higher protein intake (g/kg body weight/day) than other groups, with all groups meeting the Nordic Nutrition Recommendations [29] for dietary protein. Totals of 80%, 50%, and 25% in the UW, NW, and OW groups, respectively, were supplemented with multivitamins to reach adequate vitamin and mineral status. We did not find any differences in carbohydrates between the OW and UW groups. However, we could speculate that there is a trend of UW individuals having a lower carbohydrate intake in general, which could not be found in our data due to the low number of participants.

Hyperinsulinemia significantly elevates the risks of obesity, type 2 diabetes, and developing cardiovascular disease [33]. Within the present OW group, plasma insulin concentrations were nearly twice as high as in the other two groups. The HOMA-IR index scores were also higher in conjunction with higher body fat compared to the other groups. BDNF orchestrates numerous cellular processes underpinning the development and maintenance of normal brain function [34]. Remarkably, we observed a 40% lower plasma BDNF concentration in OW subjects, compared to both the NW and UW groups. This diminished BDNF concentration may relate to increased body weight. However, in a recent study, low plasma BDNF levels were found in children with severe CP versus typically developing controls [14]. The group of children with CP had exceptionally lower body weight (27 kg versus 40 kg for the controls), underscoring the multifactorial influences on BDNF. Notably, physical activity has proven effective in augmenting circulating BDNF levels and enhancing brain function [34].

## 5. Conclusions

Our findings emphasize that fluctuations in energy uptake, consumption, and resting metabolic rate do not predominantly account for UW status in this population. Eating challenges are the pivotal issue. Nonetheless, it is pertinent to acknowledge that subjects exhibited energy equilibrium irrespective of body weight, indicating that UW conditions were likely established prior to our study, potentially during childhood. Further exploration is imperative to unravel the supplementary factors that govern and modulate energy balance and digestive processes across the developmental phases encompassing childhood and adulthood within the population of individuals with CP.

## Figures and Tables

**Figure 1 nutrients-15-05042-f001:**
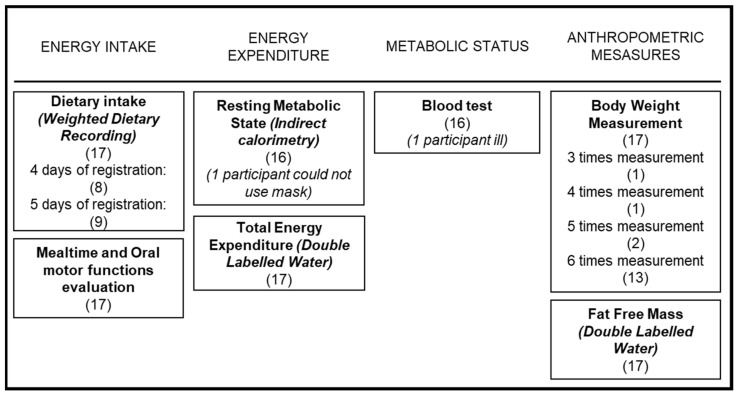
Test overview.

**Figure 2 nutrients-15-05042-f002:**
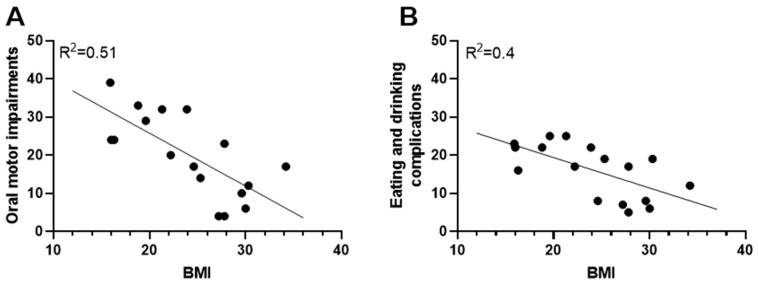
Oral motor impairments and weight status. (**A**) The relationship between BMI (body mass index) and oral motor impairments. (**B**) Eating and drinking complications. Negative regression lines are presented. Pearson correlation analysis was applied.

**Figure 3 nutrients-15-05042-f003:**
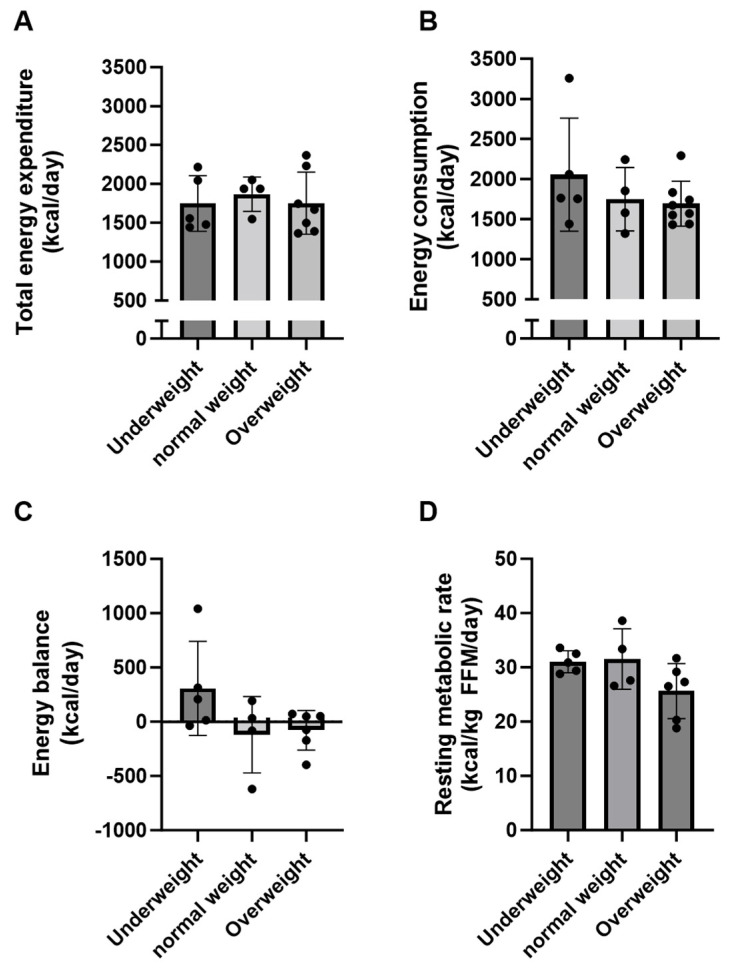
Energy balance. This figure depicts the measured factors influencing the energy balance. (**A**) Total energy expenditure (kcal/day). (**B**) Energy consumption (kcal/day). (**C**) Energy balance (kcal/day). (**D**) Resting metabolic rate related to FFM (kcal/kg FFM/day). Data are presented as means ± SD.

**Table 1 nutrients-15-05042-t001:** Subject characteristics.

	Underweight	Normal Weight	Overweight
Number of participants	5	4	8
Gender (M/F)	5/0	3/1	5/3
Age (years)	46 ± 9.6 (34–56)	47 ± 7.7 (39–56)	52 ± 21.5 (22–78)
Height (cm)	165 ± 7.6 (155–172)	166 ± 6.0 (157–170)	161 ± 8.7 (147–172)
Body weight (kg)	46.9 ± 5.3 (39.9–54.4)	63.4 ± 7.5 * (54.8–70.3)	75.7 ± 9.6 **^,(#)^ (60.1–89.7)
BMI	17.3 ± 1.7 (15.9–19.6)	23.0 ± 1.5 (21.3–24.6)	29.0 ± 2.7 (25.3–34.2)
GMFCS (I–V)			
III (n)	3	0	2
IV (n)	1	2	5
V (n)	1	2	1
Type of CP			
Spastic (n)	2	2	8
Dyskinetic (n)	3	2	0
Topographic classification			
Hemiplegia (n)	0	0	1
Diplegia (n)	1	1	3
Quadriplegia (n)	4	3	4

Characteristics of the participants divided into the subgroups of underweight, normal weight, and overweight. * *p* < 0.05, ** *p* < 0.001 vs. underweight. ^(#)^
*p* = 0.065, vs. normal weight. Mean values ± SD and range. Abbreviations: BMI: body mass index, Gender (M/F): male/female, GMFCS: Gross Motor Function Classification System.

**Table 2 nutrients-15-05042-t002:** Eating and drinking ability.

	Underweight	Normal Weight	Overweight
EDACS (I–V)			
II n, (%)	1 (20)	1 (25)	7 (87.5)
III n, (%)	0 (0)	1 (25)	0 (0)
IV n, (%)	4 (80)	2 (50)	1 (12.5)
Level of assistance required during eating			
IND n, (%)	0 (0)	1 (25)	4 (50)
RA n, (%)	1 (20)	0	1 (12.5)
TD n, (%)	4 (80)	3 (75)	3 (37.5)
Modified food texture n, (%)	4 (80)	3 (75)	1 (12.5)
Modified drink consistency n, (%)	1 (20)	2 (50)	1 (12.5)
Meal time (min)	27.4 ± 6.1 (12–45)	23.8 ± 6.5 (9–39)	16.3 ± 2.8 (8–28)

Table 2 describes the eating and drinking ability of subjects. Data are presented as mean ± SEM, and for meal time, the range is included. Abbreviations: EDACS: Eating and Drinking Ability Classification System [21], IND: independent, RA: requires assistance, TD: totally dependent.

**Table 3 nutrients-15-05042-t003:** Daily intake of energy and macronutrients.

	Underweight	Normal Weight	Overweight	RI
Energy intake (kcal)	2055 ± 316	1750 ± 198	1692 ± 99	
Macronutrients				
Protein (E%)	15.4 ± 2.1	15.0 ± 1.4	17.4 ± 0.9	10–20
Protein (g/day)	74 ± 8.8	64 ± 11.7	70 ± 4.8	-
Protein (g/kg/day)	1.58 ± 0.2	1.03 ± 0.2 ^(^*^)^	0.95 ± 0.1 *	≥0.83
Protein (g/kg FFM/day)	1.94 ± 0.2	1.45 ± 0.3	1.63 ± 0.1	-
Carbohydrates (E%)	38.1 ± 4.0	50.0 ± 2.6 *	47.2± 1.9	45–60
Carbohydrates (g/day)	175 ± 8.8	208 ± 19.6	188 ± 15.9	-
Dietary fibers (g/MJ)	1.5 ± 0.2	1.5 ± 0.1	2.7 ± 0.1 ***^,###^	>3
Fat (E%)	46.9 ± 5.6	35.7 ± 1.7	34.9 ± 1.8 *	25–40
N-3 fatty acids (E%)	1.1 ± 0.2	1.3 ± 0.1	1.2 ± 0.2	>1
Added sugar (E%)	11.2 ± 2.3	22.2 ± 2.7	10.3 ± 2.9	<10

Table 3 is the daily intake of energy and macronutrients. ^(^*^)^ *p* = 0.055, * *p* < 0.05, *** *p* < 0.001 vs. underweight. ^###^ *p* < 0.001 vs. normal weight. Data are presented as mean ± SEM. RI = recommend intake from Nordic Nutrition Recommendations [29].

**Table 4 nutrients-15-05042-t004:** Fasting plasma concentrations.

	Underweight	Normal Weight	Overweight
Glucose (mmol/L)	5.3 ± 0.2	4.8 ± 0.1	5.6 ± 0.3
	(4.9–5.8)	(4.6–5.2)	(4.8–6.4)
Insulin (µU/mL)	6.4 ± 1.5	6.9 ± 1.2	12.4 ± 2.3
	(3.1–11.9)	(4.4–9.2)	(4.6–22.7)
HOMA-IR	1.5 ± 0.3	1.5 ± 0.3	3.1 ± 0.7
	(0.7–2.6)	(0.9–1.9)	(1.2–6.5)
Triacylglycerol (mmol/L)	0.8 ± 0.1	2.1 ± 1.2	1.5 ± 0.2
	(0.6–1.2)	(0.8–5.7)	(0.6–2.5)
Total cholesterol (mmol/L)	4.5 ± 0.1	4.6 ± 0.3	4.8 ± 0.3
	(4.2–4.8)	(4.0–5.3)	(3.3–5.8)
HDL cholesterol (mmol/L)	1.4 ± 0.1	1.4 ± 0.3	1.2 ± 0.1
	(1.1–1.7)	(0.7–2.0)	(0.9–1.8)
LDL cholesterol (mmol/L)	3.0 ± 0.1	2.5 ± 0.2	3.2 ± 0.4
	(2.7–3.4)	(2.1–3.0)	(2.1–4.8)
25 (OH) vitamin D (nmol/L)	69 ± 14.4	77 ± 12.6	44 ± 5.3
	(35–121)	(45–106)	(22–56)
BDNF (pg/mL)	11,031 ± 2360 (4301–16703)	10,760 ± 1885 ^(*)^ (6723–14591)	4407 ± 1303 * (522–9733)

Table 4 indicates the plasma concentrations of the three groups. HOMA-IR: homeostasis model assessment–insulin resistance index. BDNF: brain-derived neurotrophic factor. ^(*)^ *p* = 0.07, * *p* < 0.05 vs. Underweight. Mean ± SEM (range).

## Data Availability

Data from the study will be made available on request.

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
