# Peer review of "Oral Motor Impairments Contribute to Weight Status of Adults with Severe Cerebral Palsy"

_nutrients, 2023, doi:10.3390/nu15245042_

Round 1

Reviewer 1 Report

Comments and Suggestions for Authors

The study evaluated relationship between dietary intake and motor functions assessed by professional occupational therapists during mealtimes in a group of patients suffering from various types of cerebral palsy. All patients originated from one 24 hour caretaker home in Copenhagen, Denmark. This study is novel and well written. However, the manuscript could be improved in a few places. See below:

Too many abbreviations making the text difficult to read: .  line 91 – spell out BW and throughout the text; Consider not using abbreviations of rarely used names. Abbreviations make the results section very difficult to follow.

Provide a short description of GMFCS scores for readers who are not experts in the area.

Table 1: no need to compare BMI, as the groups were defended based on BMI (body weight category).

Table 1 should be in the Results section, following the description of statistical methods.

Table 1: compare statistically frequencies of GMFCS scores, topographic classification, and type of CP between BW categories. Please add the description into the stats section.

There is a significant difference in frequencies of type of CP between BW by a Chi square test. How does it affect the results? Should this limitation be included in the discussion?

Describe in more detail the DLW method for readers not familiar with it (line ~190).

I am impressed how precisely food intake was measured (within 1g). However, it is possible that the caregivers were more involved with feeding the patients when the sessions were taped; this could be the explanation for positive energy balance in the UW group.

In the OW group, there is less participants totally dependent on their caregivers that in both UW and NW. Surprisingly, the UW group consumed more energy than the NW group, and both of them consumed more energy than the OW. Is it possible that the caregivers were giving these participants more attention than usual, given that the feeding sessions were taped? Please add discussion of this potential confound.

Line 310: is the carbs intake not different between ALL groups, as per ANOVA?

Significant proportion of OW has insulin resistance (HOMA-IR > 2.5). Please provide the exact number of cases.

Reviewer 2 Report

Comments and Suggestions for Authors

The topic of the article refers to eating difficulties in a group of adults with severe cerebral palsy (CP) who additionally suffer from oral motor impairments. This issue is important from the point of view of therapy and everyday care of these individuals. The authors assumed that "the underweight condition arises from inadequate energy intake due to difficulties in eating, rather than heightened total energy expenditure or an elevated resting metabolic rate." In order to verify this hypothesis, they conducted research in a group of 17 adults diagnosed with severe CP. The analysis of the results showed, among others, presence of significant correlation between weight, oral impairments and eating difficulties. Also, total energy expenditure and daily consumption were similar between underweight and overweight participants. Finally, the authors concluded that “eating difficulties are the central concern for underweight status in the study population”.

From a methodological point of view, the study was conducted correctly. The interpretation of the data is within the scope of the results obtained. A certain disadvantage is the relatively low number of people surveyed. For this reason, the results cannot be generalized to the entire population of CP patients. Therefore, I recommend including a "limitations" section at the end of the article. Also, the authors should mention in the methodological part that the study is a pilot study, especially since they themselves postulate the continuation of this type of research among patients from different age groups. After making these changes, I support the publication of the article.

Reviewer 3 Report

Comments and Suggestions for Authors

In this article, the authors assessed several factors that may influence the weight of adults with severe cerebral palsy. This study is very interesting and may be important for practitioners and caregivers working with patients with severe cerebral palsy.

I have a few doubts, I ask the authors to explain them.

*Has the study been registered as a clinical trial?

*Please complete the section on materials and methods, in the introduction to the research project, by specifying the duration of the entire experiment.

*Did the authors have any limitations in the study? If so, please complete the description.
